# Challenges for AI methods in Traditional Chinese Medicine

## Abstract

Applications of AI (Artificial Intelligence) in fundamental medicine greatly vary from human genetics to clinical testing, from protein structure prediction studies to electronic health records. Integrative approach to human health relies on traditional approaches of Oriental Medicine having own system of knowledge presentation, symptoms and diagnostics conceptions. We note traditional medicine approaches in Russia and Asian countries not yet formalized in computer databases, discuss current trends for data analysis and medical knowledge representation in AI.

Here we review integrative medical approaches using classical and traditional healthcare methods knowledge from point of view of AI. These tools leverage machine learning, databases, and large language models (LLMs) to handle TCM's complexity, including herbal formulations and physiological modeling. Machine learning aids in tongue image analysis (that is canonical diagnostics for TCM), pulse diagnosis, and syndrome differentiation, improving accuracy over traditional methods. We highlight approaches to analyze herbal drug components and active ingredients used.

To recap, we note series of applications of AI methods for data standardization, TSM ontology description, disease classifications using predictive tools such as neural networks, and LLM for process description and medical decision support.

## 1 Introduction

AI technology finds its extensive applications in almost every aspect of healthcare and associated fields, such as robotics-mediated complex surgical procedures, robotics in high-throughput clinical diagnosis and therapy, telemedicine, developing universal coding systems for exchange, storage, interpretation, and quick retrieval of healthcare-associated information (Chen et al., 2017; Feng et al., 2021). Molecular mechanisms of human disease progression have complex genetic underpinnings, and sophisticated sequencing approaches coupled with advanced analytics (Orlov et al., 2021). Modern computational approaches for the search and analysis of potential drug targets are based on sequencings and omics technologies being far from clinical cases. Holistic approach to human health studies has been developed for thousand years in Asia, especially in China, forming own methodological and conceptual system, own way of knowledge presentation.

We note traditions and medical knowledge systems in India, South-East Asia, Korea (Kulshreshtha et al., 2025; Kwon, 2025; Jeong and Lee, 2025; Rani et al., 2026). The term Traditional East Asian Medicine (TEAM) for the clinical decision-making is also used (Bae et al., 2025). Here we describe modern applications of AI technique to Traditional Chinese Medicine (TCM). The foundation of TCM lies in its holistic approach, manifested through herbal compatibility theory, which has emerged from extensive clinical experience and evolved into a highly refined knowledge system (Chen et al., 2025b).

AI tools are advancing Traditional Chinese Medicine (TCM) by integrating ancient practices with modern data analysis, particularly in diagnostics, drug discovery, and data integration (Ji et al., 2026; Ge et al., 2026). Chinese scholars attempted to combine AI technologies with traditional Chinese medicine (TCM) to develop an AI-guided assistive diagnostic and therapeutic system within the realm of TCM since 1970s (Bai, 2011).

Now Artificial intelligence is empowering all stages of TCM new drug development with unprecedented depth (Lu et al., 2025). These problems include: semantic understanding, reconstruction, and

dialogue of literature based on natural language processing (NLP) and large language models(LLMs); safety modeling and druggability assessment driven by statistical learning, including deep learning; syndrome objectification via multimodal learning that integrating heterogeneous data such as tongue images, pulse patterns, and electronic medical records; and intelligent optimization of clinical research through adaptive trial design, platform trials, and reinforcement learning (Tang et al., 2024).

Here we consider challenges of TCM in mechanistic understanding of syndromes and herbal formulations, novel drug discovery, and the delivery of high-quality, patient-centered clinical care (personalized medicine) (Xu et al., 2019). Traditional herbal formulas for TCM are actively studied using omics technologies (Anashkina et al., 2025). AI combines network pharmacology with multi-omics (genomics, proteomics, metabolomics) to decode polypharmacological mechanisms of herbal components, screen active compounds and predict targets for diseases like cancer or inflammation (Orlov et al., 2021; Jin et al., 2024).

Development of AI technologies is getting recognized by medical staff working with TCM. The national survey on the integration of traditional Chinese medicine and artificial intelligence in China revealed wide interest and acceptance of new technologies — about 62% of medical staff were willing to try TCM diagnosis and treatment services combined with AI (Gu et al., 2025). To complement the survey, Hu et al. (2025) used questionnaires to estimate attitude and acceptance of AI for patients — individuals with health needs, including patients seeking TCM/Western medical treatment. A cross-sectional national survey was conducted at 13 medical institutions across China. About 63% of respondents were familiar with the TCM-AI equipment and were willing to try TCM diagnosis and treatment services combined with AI. But only 43% of respondents trusted the diagnosis results provided by the TCM-AI equipment (Hu et al., 2025).

Note HERB 2.0 pharmacotranscriptomics datasets (Fang et al., 2021) map herbal effects to gene expression profiles, identifying similarities for drug repositioning. Arunachalam et al. (2025) developed SIMPD (South Indian Medicinal Plants Dataset) tool, a curated dataset comprising high-resolution images of diverse medicinal plant species native to South India, specially design for machine learning applications.

We discuss language models for disease description and success in this field. AI models physiology via systems theory frameworks for holistic TCM mechanisms and LLMs for diagnosis simulation, literature mining, and prescription generation from ancient texts and cases (in Chinese) can extract insights from TCM records, building knowledge graphs for clinical decisions. Several specific AI tools enhance TCM diagnosis by automating and objectifying traditional methods like meridian analysis, tongue inspection, and pulse reading (Chen et al., 2022).

Wang et al. (2026b) discussed network pharmacology and TCM approaches. Modern medical research paradigm of "single drug, single target" presents significant challenges due to its holistic approach. Network pharmacology and its core theory of network targets connect drugs and diseases from a holistic and systematic perspective based on biological networks, overcoming the limitations of reductionist research models and showing considerable value in TCM research. Recent integration of network target computational and experimental methods with artificial intelligence (AI) and multimodal multi-omics technologies has substantially enhanced network pharmacology methodology.

## 2 TCM DOMAIN KNOWLEDGE REPRESENTATION

We start our review with representation of knowledge of TCM domain to be formalized. The complex diagnostic and treatment model used in TCM is based on a "symptom-pattern-disease-formula" framework that heavily relies on practitioners' experience (Duan et al., 2025). However, this model faces several challenges, including ambiguous knowledge representation, unstructured data, and difficulties with knowledge sharing. Recent advancements in artificial intelligence, natural language processing, and medical knowledge engineering have significantly improved research on knowledge graphs (KGs) and intelligent diagnosis and treatment systems for these disorders, making these technologies crucial for modernizing TCM.

This article systematically reviews two core research pathways related to Spleen-Stomach disorders. The first pathway focuses on constructing knowledge graphs for "structured knowledge representation". This includes ontology modeling, entity recognition, relation extraction, graph fusion, semantic reasoning, visualization services, and an ensemble model to predict treatment efficacy. The second

pathway involves the development of intelligent diagnosis and treatment systems, with a focus on "clinical applications". This pathway includes key technologies such as quantitative modeling of TCM, the four diagnostic methods (inspection, auscultation-olfaction, interrogation, and palpation), semantic analysis of classical texts, pattern differentiation algorithms, and multimodal consultation recommenders. Through the synthesis and analysis of current research, several ongoing challenges have been identified. These include inconsistent models and annotation of TCM clinical knowledge, limited semantic reasoning capabilities, insufficient integration between KGs and intelligent diagnostic models, and limited clinical adaptability of existing intelligent diagnostic systems. To address these challenges, this review suggests future research directions that include enhancing heterogeneous multisource knowledge integration techniques, deepening semantic reasoning through collaborative reasoning frameworks that incorporate large language models, and developing effective cross-disease transfer learning strategies.

Duan et al. (2025) considered it on example of Spleen-Stomach disorders as prevalent clinical conditions in Traditional Chinese Medicine. Traditional Chinese medicine (TCM) represents a paradigmatic approach to personalized medicine, developed through the systematic accumulation and refinement of clinical empirical data over more than 2000 years, and now encompasses large-scale electronic medical records (EMR) and experimental molecular data (Yan et al., 2025).

In Traditional Chinese Medicine Electronic Medical Records (TCM EMRs), symptom descriptions are often semi-structured, and coarse-grained annotation can lead to symptom nesting and information loss. To address these limitations and improve the precision of symptom representation, Gou et al. (2025) proposed a fine-grained symptom entity annotation system. Its objective is to convert unstandardized symptom expressions into structured data, thereby enhancing the correlation and standardization of symptom information to support intelligent TCM diagnosis and treatment.

Wang et al. (2025a) presented benchmark dataset for TCM. LLM capacities to support rational medication use and guarantee prescription safety remain insufficiently investigated-especially in tasks such as prescription audit, which plays a critical role in safeguarding both. This paper presents TCMEval-PA, a benchmark dataset for assessing the capabilities of LLMs in prescription audit of Chinese herbal medicines. The dataset comprises 328 choice questions, including 297 single-choice and 31 multiple-choice. All questions were designed and compiled through rule extraction from official documents and reviewed by licensed TCM physicians. TCMEval-PA comprehensively encompasses the key dimensions of prescription safety, including normativity (e.g., dispensing, decoction requirements, and regulations for special medicines) and appropriateness (e.g., contraindicated combinations and excessive dosages) (Wang et al., 2025a).

Gou et al. (2025) annotated 500 TCM EMRs over five trials, identified 12 entity categories and 10 relation types. The inter-annotator agreement F1 scores for entities and relations were 93.5% and 91.2%, respectively.

The complex diagnostic and treatment model used in TCM is based on a "symptom-pattern-disease-formula" framework that heavily relies on practitioners' experience. However, this model faces several challenges, including ambiguous knowledge representation, unstructured data, and difficulties with knowledge sharing. Recent advancements in artificial intelligence, natural language processing, and medical knowledge engineering have significantly improved research on knowledge graphs (KGs) and intelligent diagnosis and treatment systems for these disorders, making these technologies crucial for modernizing TCM. Duan et al. (2025) systematically reviewed two core research pathways related to Spleen-Stomach disorders.

Application of AI for pharmacopuncture as presented by (Kwon, 2025). Pharmacopuncture is a therapeutic modality used in Korean medicine that involves the injection of medicinal extracts into acupoints. This study aimed to develop an artificial intelligence (AI)-based automated system for building and maintaining a living evidence map in the field of pharmacopuncture research and verify its performance. A web-based system that automates literature search, selection, data extraction, and classification using PubMed API and Gemini AI was developed.

## 3 LARGE LANGUAGE MODELS FOR TRADITIONAL CHINESE MEDICINE

Large language models (LLMs) show promise for supporting Traditional Chinese Medicine (TCM) practice, but their clinical utility is limited by domain-specific knowledge gaps, hallucinations, and weak multi-turn reasoning (Wang et al., 2026a).

Studying the association of gene function, diseases, and regulatory gene network reconstruction demands data compatibility (Veljković et al., 2023). Data preparation and standardization challenge TCM domain.

Large language models struggle with dynamic clinical workflows and personalized treatment in complex systems like TCM. Note ChatGPT-4.0 and the ChatGLM series as novel conversational large language models (LLMs).

To improve LLM performance in specialized science domains, researchers have explored various optimization strategies. One prominent method is retrieval-augmented generation (RAG), which integrates retrieval mechanisms with generative models. This approach allows for the customization of retrieval strategies and the integration of domain-specific knowledge. For example, Clinfo.ai is a GPT-based RAG implementation that retrieves abstracts from PubMed. RAG not only enhances specialization but also reduces hallucinations and enables dynamic updates to keep pace with rapidly evolving fields. This technique has shown success in numerous clinical applications.

The current technologies remain limited when applied to domains like TCM gastroenterology. Existing general-purpose LLMs rarely encode TCM-specific concepts such as syndrome differentiation, pattern–symptom mapping, and classical formula theory, and they lack access to curated TCM gastroenterology corpora.

The AcuKG tool integrates data on acupuncture from multiple sources, including online resources, guidelines, PubMed literature, ClinicalTrials.gov, and multiple ontologies (SNOMED CT, UBERON, and MeSH) (Li et al., 2025b).

Recently Wang et al. (2025e) estimated performance of ChatGPT-like models in Traditional Chinese Medicine for metabolic associated fatty liver disease.

In the evaluation module of "Ability in syndrome differentiation and treatment principles," the performance ranking of the 4 models tested was ChatGLM4+ Knowledge Base (Wang et al., 2025e). Pretraining LLMs with TCM-specific knowledge bases while maintaining internet search capabilities substantially enhanced their diagnostic and therapeutic performance in TCM applications. Importantly, general-purpose LLMs required both domain-specific medical fine-tuning and culturally sensitive adaptation to meet the rigorous standards of TCM clinical practice.

Long et al. (2025) used empirical evaluation of different LLM types in the specialized domain of TCM stroke. Wang et al. (2026a) presented GastroTCM, a specialized LLM assistant for TCM gastroenterology built by fine-tuning a Llama3-8B model and augmenting it with a Retrieval-Augmented Generation (RAG) and an agent framework (Touvron et al., 2023; OpenAI, 2023).

In the domain of Chinese clinical medical question-answering, traditional Large Language Models (LLMs) encounter challenges such as hallucinations and difficulties in updating knowledge for knowledge-intensive tasks. To address these issues, Zhang et al. (2025) presented a Chinese clinical medical QA model that integrates Retrieval-Augmented Generation (RAG) and a medical knowledge graph, named CMedRAGBot. First, a Chinese medical knowledge graph encompassing multiple entity types-including diseases, medications, and symptoms-is constructed. Based on this knowledge graph, a Named Entity Recognition (NER) model built on a Chinese-RoBERTa and BiGRU architecture is designed, with data augmentation strategies employed to enhance its generalization capability. In addition, prompt engineering techniques are used to implement intent recognition for user queries, mapping them to predefined intent categories. Finally, the aforementioned modules are integrated to form a complete Chinese clinical medical QA system. In the experimental evaluation, CMedRAGBot is deployed on five state-of-the-art LLMs (including ChatGPT-4o, ChatGPT-o1, DeepSeek-R1, Llama-3.3-70B-Instruct, and Gemini 2.0 Flash) and tested using specialized question banks derived from the Chinese Clinical Medical Qualification Examinations (Source code of the research is available at `https://github.com/zhdongfang/CMedRAGBot`).

Traditional Chinese medicine with knowledge-intensive framework poses unique challenges to performance for large language models (Li et al., 2025c).

Qin et al. (2025) constructed RAG-CPMF, an intelligent CPM recommendation framework integrating large language models (LLMs), retrieval-augmented generation (RAG), and the largest Chinese patent medicines dataset. The accuracy of RAG-CPMF was evaluated against clinical guidelines, demonstrating that this framework significantly improved CPM recommendation accuracy compared with general-purpose LLMs (Qin et al., 2025).

Overall, the performance of existing LLMs in TCM-specific tasks remains limited due to the lack of optimization for TCM knowledge during the pre-training phase, insufficient datasets, and the constraints of fine-tuning techniques (Tong et al., 2025).

## 4 ACUPUNCTURE AND AI APPLICATIONS

We consider AI applications for oriental diagnostics classification systems such as acupuncture and body parameters (pulse, tongue color and other estimates). Acupuncture, a key modality in traditional Chinese medicine, is gaining global recognition as a complementary therapy and a subject of increasing scientific interest (Yoon et al., 2025). However, fragmented and unstructured acupuncture knowledge spread across diverse sources poses challenges for semantic retrieval, reasoning, and in-depth analysis.

Kim et al. (2026) investigated the efficiency of an AI model in predicting the acupoints and compared its performance to placements made by a practitioner of traditional Korean medicine using ear images. The mask region-based convolutional neural network (Mask R-CNN) model was utilized to isolate the ear region, followed by landmark detection using a CNN model trained on resized images to predict three auricular acupoints. The AI-driven approach showed significant potential in improving both the accuracy and consistency of auricular acupoint identification (Kim et al., 2026).

Wang et al. (2025d) discussed acupuncture, a nonpharmacological therapeutic method in relation to Alzheimer's disease (AD) treatment that has received widespread attention. With the rapid development of modern science and technology, the mechanism of action of acupuncture in the treatment of AD has gradually become increasingly clear.

AI applications in TCM include diagnostic systems like Medical's AI meridian diagnostic tool, which analyzes electrical resistance from 80 acupoints for objective diagnostics (Wang et al., 2025c).

Li et al. (2025b) developed AcuKG, a comprehensive knowledge graph that systematically organizes acupuncture-related knowledge to support sharing, discovery, and artificial intelligence-driven innovation in the field.

Acupuncture can improve cognitive function in AD patients through various mechanisms, such as reducing $\beta$-amyloid deposition, inhibiting Tau protein hyperphosphorylation, and attenuating neuroinflammation, and shows good therapeutic potential (Wang et al., 2025d).

Yoon et al. (2025) assessed the ability of GPT-4 to make medical decisions regarding acupuncture treatment by comparing its selection of acupoints with those made by human clinicians.

## 5 ANALYSIS OF PHYSIOLOGICAL PATTERNS BY AI TOOLS

Analysis of fine tune physiological parameters of human organs is important for clinical medicine (Al-Zamil et al., 2025; Artamonov et al., 2025). Pulse diagnosis holds a pivotal role in traditional Chinese medicine diagnostics, with pulse characteristics serving as one of the critical bases for its assessment (Li et al., 2025a). Accurate classification of the pulse pattern is paramount for the objectification of TCM (Chen et al., 2022).

Li et al. (2025a) used a multi-channel lightweight graph convolutional network (GCN) for classification of the pulse pattern in TCM. The proposed network model achieved 91% accuracy, a mean F1 score of 92%, a mean recall rate of 92%, and a mean precision rate of 92% on the pulse dataset.

Tongue diagnosis is the kernel method of Traditional Chinese Medicine (TCM), and it has been proved that the condition of the tongue can serve as an indicator of a person's health status. Tongue

Image Segmentation is an essential task, as the tongue is sensitive to the physiological conditions and pathological changes of patients and can help physicians determine strategies for the syndrome differentiation. To automatically recognize a person's latent diseases by computer vision technology, getting the tongue segmentation from a picture with high precision has significant importance (Tang et al., 2024). Cai et al. (2024) developed TSRNet system for Tongue image segmentation based on an encoder–decoder framework with global and local refinement. Yao et al. (2025) analyzed tongue segmentation images for TCM diagnostics using neural networks. The proposed post-processing image analysis method can effectively improve all classic neural networks in tongue segmentation (Yao et al., 2025).

Ge et al. (2026) considered myocardial ischaemia–reperfusion injury outcome combining AI tools for analysis key factors influencing postoperative mortality using TCM classification. The authors incorporated traditional Chinese medicine (TCM) classifications to reflect overall patient status, construct an optimal machine learning model for precise prognosis assessment.

Traditional Korean Medicine has own traditional diagnosis scheme. Temperature sensitivity has gained considerable attention (Jeong and Lee, 2025). This trait has long been used to identify cold-heat patterns (C-HPs), a diagnostic framework in Traditional Korean Medicine that categorizes individuals based on their thermal responses. C-HP helps understand an individual's inherent physical characteristics, which have been shown to be highly heritable and thus shaped by genetic factors. The authors incorporated genetic studies related to traits such as "Cold" or "Heat," as well as thyroid hormone, which plays a key role in thermogenesis through the activation of metabolic pathways. Set of SNP (single nucleotide polymorphisms) was found in genetic databases (Jeong and Lee, 2025).

The traditional Indian system of medicine promotes overall health and wellness through personalized therapies and a detoxifying process (Trikamji, 1941). Therapeutic emesis, or vamana karma, is one of the bio-pentavalent purification procedures used to treat deranged kapha ailments that include metabolic, respiratory, and dermatological conditions such as psoriasis and eczema. Rani and colleagues (Rani et al., 2026) presented one of the first attempts to apply deep learning for objective analysis of the therapeutic emesis process in Ayurveda. By combining YOLOv9 for vomit detection and residual neural network for classification, the framework achieves promising accuracy in automated vomit identification. The results will demonstrate the potential of AI-assisted analysis in traditional medicine (Rani et al., 2026).

## 6 CHINESE HERBAL MEDICINE APPLICATIONS

Traditional Chinese medicine formula (herbal formula) represents a fundamental component of Chinese medical practice (Chen et al., 2025b). Within this framework, Chinese herbal medicines exhibit intricated characteristics, including multi-component interactions, diverse target sites, and varied biological pathways. These complexities pose significant challenges for understanding their molecular mechanisms.

Herbal medicine has historical roots in many countries. Traditional medicinal systems such as Ayurveda offer a rich repository of plant-based remedies for inflammatory conditions (Kulshreshtha et al., 2025). In Serbia, traditional medicine, based on the strong belief in the power of medicinal herbs, is the widespread form of treatment (Radovanović et al., 2023). Kulshreshtha et al. (2025) screened 18 medicinal plants for anti-inflammatory potential using in vitro and in vivo assays.

Traditional Chinese Medicine (TCM) has long been regarded as a valuable resource for modern drug discovery. However, the limited availability of recorded entities and information, the complexity and sparsity of the herb-ingredient-target-disease network, and inconsistencies in data representation hinder the effectiveness of high-throughput screening approaches (Chen et al., 2025a).

Chen et al. (2025a) developed a data-driven and deep learning-based workflow, TCM-navigator, which enables the in-silico generation, quality control, and physics-based evaluation of TCM-like molecules.

Jia et al. (2026) analyzed current applications and limitations of AI in the image identification, quality control, active ingredient and toxicity assessment, and origin identification of Chinese herbal medicine. A comprehensive literature search was conducted in PubMed, Google Scholar, and CNKI to identify studies on the application of AI in TCM, covering image recognition, quality control, origin identification, phytochemical analysis, and toxicity assessment. The authors show

that AI offers significant advantages in the identification of herbal medical components, improving both accuracy and efficiency. In quality control, the combination of AI with spectroscopic and sensory detection enables more objective analyses. When integrated with chromatographic and multi-technique approaches, AI supports the evaluation of complex components and toxicity (Jia et al., 2026).

# 7 NETWORK PHARMACOLOGY

Network pharmacology has gained widespread application in drug discovery using mathematical approaches for network reconstruction, theory of graphs, computer methods. We note series of online tools such as GeneMANIA, STRING, PathBanks (Franz et al., 2018; Wishart et al., 2024; Szklarczyk et al., 2023) for gene network reconstruction, gene target – drug interaction networks. Associate network design was developed in the ANDsystem developed in Russia (Demenkov et al., 2011; Ivanisenko et al., 2020).

Network approach is particularly important in traditional Chinese medicine research, which is characterized by its "multi-component, multi-target, and multi-pathway" nature (Shao et al., 2025). Through the integration of network biology, TCM network pharmacology enables systematic evaluation of therapeutic efficacy and detailed elucidation of action mechanisms.

Shao et al. (2025) describes the methodology of TCM network pharmacology, encompassing ingredient identification, network construction, network analysis, and experimental validation.

AI techniques help in uncovering molecular mechanisms of drug action, its role in compound absorption, distribution, metabolism, and excretion (ADME) prediction, molecular target identification, compound and target synergy recognition, pharmacological mechanisms exploration. It helps with herbal formula optimization as well (Oborotov et al., 2023).

TCM complex multi-component compositions and intricate mechanisms of action pose significant challenges for modern scientific investigation. Addressing these complexities requires advanced techniques capable of dissecting cellular and molecular interactions with high resolution. Single-cell omics enables high-throughput, unbiased profiling of genomic, transcriptomic, proteomic, and metabolomic landscapes at single-cell resolution (Jiang et al., 2025). By identifying active constituents, pinpointing therapeutic targets, and elucidating mechanisms of action, single cell omics offer profound insights into the pharmacological and therapeutic properties of TCMs.

# 8 DISCUSSION

TCM is vital component of healthcare systems in China and worldwide, has been increasingly utilized in clinical practice. However, the problems of such treatment accept include misunderstanding, lack of explainability and trust, that interplays with traditions of technical education (Li et al., 2022). Mulugeta et al. (2024) reviewed deep learning for medicinal plant species classification and recognition. The lack of a globally available and public dataset need for medicinal plants indigenous to a specific country and the trustworthiness of the deep learning approach for the classification and recognition of medicinal plants was underlined. It indicates on difference in traditional medicine domain descriptions in world, not only in relation to TCM.

The integration of TCM and AI demonstrates promising acceptance among health-seeking individuals in China, with younger and educated populations who have health demands for TCM showing particularly high trust, and intelligent syndrome differentiation systems highlight a clear pathway for AI to modernize TCM practice by augmenting diagnostic accuracy and treatment personalization (Gu et al., 2025; Shin et al., 2025).

Traditional Chinese medicine adaptation for researchers and patients has been significantly evolved (Yang et al., 2025). TCM education policies in China include medical education, curriculum reform, rural health care, internationalization, and the integration of TCM with modern education systems. Despite importance of Chinese patent medicines (CPMs), approximately 70% of CPMs are prescribed by Western medicine physicians who lack expertise in traditional Chinese medicine syndrome differentiation and treatment (Qin et al., 2025). Studying current trends in education policies (Yang et al., 2025) revealed 5 stages of TCM policy evolution in China: marginalization, standardization, specialization, systematization, and restandardization.

According to Gu et al. (2025), the top three most important factors in the application of AI in TCM were accuracy (78.0%), convenience of operation (67.5%), and participation of medical staff (60.9%). Future research on AI in TCM diagnosis and treatment may emphasize building large-scale, high-quality TCM datasets with unified criteria based on syndrome elements; identifying algorithms suited to TCM theoretical data distributions; and leveraging AI multimodal fusion and ensemble learning techniques for diverse raw features, such as images, text, and manually processed structured data (Wang et al., 2025b).

The top three important processes of integration of TCM and AI were medical research, personalized generation of regimen, and intelligent inquiry. The top three concerns about the potential risks associated with the integration of TCM and AI were the misinterpretation of cultural contexts, flexibility in dialectical treatment, and simplification of traditional TCM experience by algorithms (Hu et al., 2025).

Despite problems of the knowledge translation for TCM systematic data analysis in Western medicine is far from being perfect. We'd note high rate wrong and unsafe responses for public medical chatbots in English. Draelos et al. (2026) described the physician-led study comparing the safety of four publicly available chatbots — Claude by Anthropic, Gemini by Google, GPT-4o by OpenAI, and Llama-3.0/3.1-70B by Meta — on a new dataset, HealthAdvice. The rate of problematic responses varied from 21% to 43%, with unsafe responses varying from 5% to 13%.

Li et al. (2025c) demonstrated that although large-scale LLMs exhibit strong knowledge recall in TCM, their suboptimal performance on multiple-choice questions and substantial computational costs may limit their practical applicability in clinical settings.

## 9 CONCLUSION

TCM is getting widespread acceptance in clinical practice. The integration of TCM and AI has promising future, prioritizing diagnostic accuracy while addressing cultural/clinical adaptation challenges in key applications, such as syndrome differentiation systems.

Although large language models (LLMs) have witnessed rapid development in medical applications, their capacities to support rational medication use and guarantee prescription safety remain insufficiently investigated-especially in tasks such as prescription audit (Wang et al., 2025a).

## ACKNOWLEDGMENTS

The authors are grateful to the colleagues from Shanghai University of Traditional Chinese Medicine for fruitful discussion. The study was supported by RSF.

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
