# OpenReview forum: "Challenges for AI methods in Traditional Chinese Medicine"
_mathai.club/MathAI/2026/Conference — MathAI 2026 Conference Submission_

### Official Review · Reviewer_CY9n · 2026-03-12
**The article is billed as a review of the application of artificial intelligence methods in traditional Chinese medicine (TCM). The article does not formulate a single clear research question. What exactly do the authors want to find out? The title does not match the content.**

**Rating:** 2
**Confidence:** 5

**Review:**

The article comes across as a hastily compiled text without a clear statement of the problem, without a review or analysis methodology, and without any original scientific contribution. It is essentially an annotated bibliography.
The article's structure is inconsistent, and the text is repetitive.

The only strength of the article is that the authors have collected over 60 references covering various applications of AI in TCM. However, a significant portion of the references are indirectly related to the stated topic.
And most importantly, it is unclear what contribution this work makes to the field of mathematical foundations of AI.

The text contains numerous stylistic and grammatical problems. Several sentences reproduce the abstracts of cited papers verbatim or nearly verbatim, raising questions about the originality of the text.

For a review article covering dozens of papers, the absence of comparative tables, method taxonomies, or even a single diagram is a weakness.

Conclusion

The article is not relevant to the MathAI conference topic. A complete lack of mathematical content means that the article cannot be published on a platform focused on mathematical methods in AI.

---

### Official Review · Reviewer_jf1G · 2026-03-12
**This paper presents a broad review of how artificial intelligence is being applied in Traditional Chinese Medicine, with emphasis on knowledge representation, large language models, acupuncture-related applications, physiological pattern analysis such as pulse and tongue diagnosis, herbal medicine analysis, and network pharmacology. The paper’s main goal is to argue that AI can help formalize, modernize, and scale TCM practice, while also highlighting ongoing challenges related to data quality, trust, explainability, and domain-specific adaptation.**

**Rating:** 5
**Confidence:** 3

**Review:**

The paper addresses a timely and interesting topic at the intersection of AI, medicine, and traditional knowledge systems. Its main contribution is as a narrative review that gathers a fairly wide range of recent studies across multiple subareas of TCM, including knowledge graphs, retrieval-augmented language models, tongue and pulse analysis, herbal compound analysis, and network pharmacology. This breadth makes the paper useful as an introductory survey for readers who want a general picture of the current landscape.


Strengths:

The paper addresses a timely and interdisciplinary topic and provides a broad overview of recent AI applications in TCM. It covers multiple relevant subareas, including knowledge graphs, LLMs, diagnostic imaging, herbal medicine analysis, and network pharmacology, and it also acknowledges important limitations such as hallucinations, trust, and the need for better datasets.

Weaknesses:

The main weakness is the lack of a clear review methodology or evaluation framework, which makes the paper read more like a descriptive literature compilation than a critical survey. The scope is also somewhat diffuse, since the discussion extends beyond TCM into other traditional medicine systems, and the manuscript would benefit from stronger synthesis, tighter focus, and substantial language polishing.

Recommendations:

The paper would be improved by adding a transparent review methodology, narrowing the scope more clearly to TCM, organizing the cited work into a stronger comparative framework, and revising the writing for clarity and coherence. A summary table contrasting methods, datasets, evaluation settings, and limitations would also strengthen the contribution.